# Therapeutic Targets in Advanced Penile Cancer: From Bench to Bedside

**DOI:** 10.3390/cancers16112086

**Published:** 2024-05-30

**Authors:** Lance C. Pagliaro, Burak Tekin, Sounak Gupta, Loren Herrera Hernandez

**Affiliations:** 1Department of Oncology, Division of Medical Oncology, Mayo Clinic, Rochester, MN 55905, USA; 2Department of Laboratory Medicine and Pathology, Mayo Clinic, Rochester, MN 55905, USA; tekin.burak@mayo.edu (B.T.); gupta.sounak@mayo.edu (S.G.); herrerahernandez.loren@mayo.edu (L.H.H.)

**Keywords:** penile cancer, HPV, immune checkpoint inhibitor, EGFR, nectin-4, antibody-drug conjugate

## Abstract

**Simple Summary:**

Cancer of the penis is a relatively uncommon cancer that nevertheless can be life threatening when it is at an advanced stage. We summarized the published laboratory results that might lead to new and better treatment approaches, including those that harness the immune system. We identified several opportunities for drug development and corresponding clinical trials that are or will soon be testing these ideas in patients. In conclusion, we outline how these ongoing efforts will hopefully improve outcomes for patients with advanced penile cancer.

**Abstract:**

Discovery of effective systemic therapies for patients with advanced penile cancer has been slow to occur. Comprehensive genomic profiling from several studies shed light on the molecular oncogenesis of penile squamous cell carcinoma (PSCC) and differences between HPV-related and unrelated tumors. While these two subsets of PSCC appear distinct in their biology, there are not yet specific treatment strategies recommended on that basis. Cell surface proteins have been identified that may potentially serve as drug targets for monoclonal antibodies or small molecule inhibitors. Here, we review some of the new biological insights regarding PSCC that could lead to improved therapies, as well as the related clinical trials recently completed or in progress. We conclude that antibody-drug conjugates are especially promising, as are the combinations of immune checkpoint inhibitors with other types of drugs.

## 1. Introduction

Penile cancer is a rare disease in the United States, as it is in most Western/industrialized countries, while being more common in lower income countries around the world [1,2]. Discovery of effective systemic therapies for patients with advanced penile cancer has been slow to occur. The current standard treatment for unresectable or metastatic disease is cisplatin-based chemotherapy or chemo-radiotherapy. The prospective study of safer and more effective chemotherapy regimens has benefitted some patients in need of first-line or neoadjuvant treatment [3,4], but there remains a glaring unmet need for novel systemic therapies that are both more effective and better tolerated. In comparison to other more common solid tumors, the impact of targeted and immune-based therapy for penile cancer remains small.

Approximately 30–50% of patients diagnosed with penile cancer have evidence of prior infection with human papilloma virus (HPV) [5,6]. Analysis of tumoral RNA reveals evidence of viral gene expression in (high-risk) HPV-related cases, as well as overexpression of p16, a protein associated with viral oncoprotein effects on cell cycle regulation. In concert with these observations, HPV-unrelated penile cancer is distinguished by differences in genomic landscape, including more frequent mutation or loss of TP53 and a slightly worse prognosis [7,8]. Such observations hint at subsets of patients who might benefit from specific therapies, as well as opportunities to extrapolate from other HPV-dependent cancers.

In this review, we summarize the biological insights that are emerging from preclinical studies, their potential significance for development of clinical interventions, and the status of related clinical trials for penile cancer.

## 2. Molecular Profile

Comprehensive genomic profiling from several studies has shed light on the molecular oncogenesis of penile squamous cell carcinoma (PSCC), differences between HPV-related and unrelated tumors, and similarities with head and neck squamous cell carcinoma [9]. Molecular pathways that are potential targets for therapy include DNA damage response, PI3K/AKT/mTOR, and tyrosine kinase receptor pathways.

Genes with commonly reported mutations across multiple studies include *TP53*, *CDKN2A*, *NOTCH1*, and *PIK3CA* (Table 1). Certain mutations, including those in *TP53*, *FAT1*, *CASP8*, and *FBXW7*, are more common in HPV-negative than HPV-positive PSCC [7,10].

Programmed death-ligand-1 (PD-L1) immunohistochemistry is positive in 33–60% of both HPV-related and -unrelated tumors [7,14,15,16]. High tumor mutational burden (TMB) is seen in 18–20% of cases, with one study finding TMB of at least 10 mutations per Mb in 30.8% of HPV-related tumors, but not in HPV-unrelated tumors [7,9]. Both PD-L1 expression and TMB have relevance for possibly guiding treatment of PSCC with immune checkpoint inhibitors (ICIs). Mismatch repair deficient or microsatellite instability-high status are rare in PSCC, and seldom constitute an indication for immunotherapy [17,18].

### 2.1. TP53 and CDKN2A

The reported overall incidence of *TP53* mutations in PSCC is 15–58%, and of *CDKN2A* is 23–54%. Both are seen almost exclusively in HPV-unrelated tumors, suggesting differences in oncogenesis when HPV viral oncoproteins are present or absent [7]. While these two subsets of PSCC appear distinct in their biology, there are not yet specific treatment strategies to recommend on that basis. 

The HPV viral proteins E6 and E7 block the DNA damage response and cell cycle regulation by their interaction with TP53 and retinoblastoma (RB) protein, respectively. Therapeutic vaccine strategies for HPV-related cancers have focused, in part, on the E6/E7 oncoproteins, as discussed in the section on combinations with immunotherapy below.

### 2.2. NOTCH1 (Notch Pathway)

Mutations involving *NOTCH1* are seen in both HPV-related and -unrelated PSCC, with an overall frequency of 14–35% reported [12,19]. One study found that the Notch pathway was altered in 70.6% of cases [9]. *NOTCH1* is frequently mutated in head and neck cancer, where it functions as a tumor suppressor gene, as it appears to be for squamous cell carcinomas from other sites such as PSCC [20]. In other cancer types, however, *NOTCH1* is thought to function as an oncogene [21]. 

The Notch pathway is remarkably simple, where the Notch protein is a membrane receptor that, when activated, translocates to the nucleus, without intermediary signal transduction proteins. The ligands for Notch are also surface proteins, such that signaling occurs during contact with neighboring cells. Preclinical and limited clinical data suggest that downregulation in the Notch pathway may render cells sensitive to PI3K/mTOR inhibition [10]. Other genes in the PI3K/mTOR pathway that are mutated in PSCC, although less commonly, include *NF1* and *PTEN* [11].

### 2.3. PIK3CA and EGFR (Hippo Pathway)

The Hippo pathway is a key oncogenic driver in multiple tumor types. Impaired regulation of genes controlled by the Hippo downstream transcriptional coactivators YAP (Yes-associated protein 1) and TAZ (WWTR1, WW domain containing transcription regulator 1) occurs, for example, in head and neck squamous cell carcinoma [22]. Interestingly, YAP/TAZ signaling might not be as important for normal homeostasis as it is for oncogenic growth. 

*PIK3CA* and epidermal growth factor receptor (*EGFR*) mutations can dysregulate the Hippo signaling pathway in cells [23]. Recent advances in the role of the Hippo pathway in head and neck cancer have provided evidence that genetic alterations of *PIK3CA* or functional loss of FAT1 (FAT atypical cadherin 1) can result in YAP activation [22]. *PIK3CA* mutations are common in PSCC, reported in 20–25% of cases. One study reported that they were more common in HPV-related (30.8%) than -unrelated (6.8%) tumors [7], although other studies found no such association with HPV. 

*EGFR* gene mutations are rarely detected in PSCC, whereas one study found *EGFR* gene amplification present in 7.8% of cases [7,11,24,25]. EGFR protein expression detected by immunohistochemistry is very common in PSCC, even in the absence of genetic alterations, suggesting that it could be a viable target for therapy [13].

## 3. Clinical Activity of Immune Checkpoint Inhibitors

There is so far insufficient evidence to support PD-L1 as a predictive biomarker in PSCC [26]. High tumor mutation burden (TMB), on the other hand, is predictive across multiple tumor types [27]. Pembrolizumab, an anti-PD-1 ICI, is approved in the United States for tumors with TMB of 10 mutations per Mb or higher, regardless of primary site. Up to 20% of patients with PSCC (and possibly a higher percentage of HPV-related tumors) may be eligible for pembrolizumab based on high TMB.

Much of the data regarding efficacy of ICI treatment for PSCC have come from case reports, retrospective studies, and basket trials. One notable exception, the phase 2 PERICLES trial, studied the efficacy of atezolizumab (anti-PD-L1) in patients with metastatic or unresectable PSCC with or without radiotherapy [17]. All patients (N = 32) received atezolizumab (1200 mg) once every 3 weeks. Twenty patients who were expected to benefit from radiotherapy to locoregional disease received additional irradiation as part of the treatment. One-year PFS was 12.5% (95% CI, 5.0 to 31.3), which was the study’s primary end point, and this did not meet the target PFS of at least 35%. Median OS was 11.3 months (95% CI, 5.5 to 18.7). In the objective response-evaluable population, the overall response rate (ORR) was 16.7% (95% CI, 6 to 35), including two (6.7%) complete responses and three (10%) partial responses. There were no statistically significant differences in outcome based on PD-L1 expression. TMB and MSI status were not characterized. High-risk HPV and p16 were both prognostic for better PFS (median 5.3 versus 2.6 months). While a link between HPV status and response to atezolizumab was not established, an exploratory analysis of CD3^+^CD8^+^ T-cell infiltration found significantly greater PFS with high versus low infiltration, suggesting an immune mechanism was relevant. 

Cemiplimab is an anti-PD1 monoclonal antibody that was studied in cutaneous squamous cell carcinoma, leading to approval for that indication in the United States [28]. For recurrent or metastatic cervical cancer after first-line platinum-based therapy, the EMPOWER-Cervical1 trial randomized patients to single-agent cemiplimab or investigator choice single-agent chemotherapy. Median OS was 12 months with cemiplimab and 8.8 months with chemotherapy [29]. Some retrospective data on cemiplimab in patients with PSCC or vulvar squamous cell carcinoma have suggested a benefit in these cancers as well [30]. Case reports of PSCC treated with cemiplimab have described objective responses, including complete responses [31,32]. In the ongoing EPIC trial, patients with previously untreated PSCC are treated with cemiplimab alone or in combination with a standard cisplatin-based chemotherapy regimen. It is a non-randomized phase 2 study in which patients are assigned to one of two arms based on their ability to tolerate chemotherapy. The endpoint for both arms is the clinical benefit rate [33].

In a phase 2 study of ipilimumab and nivolumab for rare genitourinary tumors, there were five patients with advanced PSCC, of which two (40%) had stable disease [34].

The Global Society of Rare Genitourinary Tumors recently published a retrospective study of ICIs for PSCC by collating patient data from 24 institutions [26]. Of 92 patients, 57 (62%) were treated with a single-agent anti-PD1 checkpoint inhibitor (pembrolizumab, nivolumab, or cemiplimab), 11 patients (12%) received ipilimumab and nivolumab, and the remaining 24 (26%) received other checkpoint inhibitors or drug combinations. Eighty-five patients were evaluable for response, of which there were two complete responses and nine partial responses for ORR of 13%. There were 28% with stable disease, for a disease control rate of 41%. The median TMB was 9.7 (IQR 4-12) based on 18 patients for whom data were available. All the 32 patients tested for microsatellite instability were microsatellite stable. There were no differences in outcome based on high-risk HPV status or PD-L1 immunohistochemistry.

In summary, the benefits of ICI monotherapy for patients with advanced PSCC appear to be limited based on the available data. It remains to be determined whether ICIs used in multidrug combinations will yield better results.

## 4. Therapeutic Targeting of EGFR Pathway

The EGFR pathway is implicated as potentially targetable in PSCC on the basis of *EGFR* gene amplification, protein levels by immunohistochemistry, and alterations in Hippo pathway signaling. Both *EGFR* pathway inhibitors and anti-EGFR monoclonal antibodies are of interest for potential use in PSCC (Table 2). 

### 4.1. Clinical Experience with Receptor Inhibitors

Dacomitinib is a potent, irreversible pan-human epidermal receptor (HER) inhibitor approved in the US for the treatment of non-small-cell lung cancer with *EGFR* exon 19 deletion or exon 21 p.L858R substitution alterations. A single-arm study of dacomitinib monotherapy was performed in PSCC patients (N = 28) with regional lymph node or distant metastases and no prior systemic therapy [38]. Patients were unselected with respect to EGFR expression, gene amplification, or gene mutations. The overall response rate was 32.1% (complete response rate 3.6%). Median PFS and OS were 4.1 months and 13.7 months, respectively. Dacomitinib was well tolerated, with Grade 3 skin toxicity observed in 10.7%. Molecular analysis of tumors from 25 patients revealed *EGFR* gene amplification in four patients (16%), of which two had response to treatment. There were no activating mutations of the *EGFR* gene, however, which is consistent with prior studies. Interestingly, mutations involving other PI3K/AKT/mTOR pathway genes were present in 42.9% of responders and 8.3% of non-responders. 

These results demonstrate modest clinical activity of dacomitinib in unselected patients, and perhaps more promising prospects for biomarker-selected subgroups based on *EGFR* gene amplification or activation of the Hippo pathway. Dacomitinib as a salvage treatment after chemotherapy has not yet been explored. Case reports with EGFR inhibitors erlotinib or gefitinib in later lines of treatment for advanced PSCC were not encouraging [36].

### 4.2. Clinical Experience with Anti-EGFR Monoclonal Antibodies

A retrospective study of the anti-EGFR monoclonal antibody cetuximab administered to 17 patients alone or in combination with cisplatin demonstrated four partial responses (23.5%), including 2 patients with apparently chemotherapy-resistant tumors [36]. A prospective study of panitumumab monotherapy included 11 patients with at least one prior line of systemic therapy, of which two patients had a complete response and one patient had a partial response (ORR 27%) [37]. A combined analysis of several published case reports and case series included eight patients treated with cetuximab, panitumumab, or nimotuzumab alone, and twenty patients treated with a combination of chemotherapy plus a monoclonal antibody. Overall, 50% showed a response to treatment, with a median PFS of approximately 3 months [24]. These results indicate a possible benefit in terms of palliation for patients with advanced PSCC, albeit with a short duration of response.

### 4.3. Combination Anti-EGFR Monoclonal Antibody with Checkpoint Inhibitor

Recent clinical reports point to the efficacy of a combination of anti-EGFR and anti-PD(L)-1 monoclonal antibodies in non-penile cutaneous squamous cell carcinoma, head/neck squamous cell carcinoma, and colorectal cancer [39]. Anti-EGFR antibody-driven immune activities include antibody-dependent cell-mediated cytotoxicity (ADCC), whereby immune cells can target and kill antibody-coated cells, and T-cell priming. Preclinical studies have supported these mechanisms and potential synergy with immunotherapy in a wide range of solid tumors. Considering that immune checkpoint inhibitors and anti-EGFR targeted therapies have limited clinical efficacy in PSCC by themselves, the possibility of combination with immunotherapy warrants further study.

To our knowledge, the anti-EGFR plus ICI combination has not been tested prospectively in patients with PSCC. In one retrospective analysis, a single-institution series of 17 patients with advanced PSCC received a complex regimen of chemotherapy (paclitaxel, ifosfamide, and cisplatin) plus nimotuzumab (anti-EGFR) and toripalimab (anti-PD1) [40]. Patients received up to six cycles. Partial responses were seen in 13 patients (76%). While there were no clinical complete responses, twelve patients with locally advanced PSCC went on to surgery, at which nine were pN0 and six had pathologic complete response. A phase 2 neoadjuvant study is planned based on these preliminary results (NCT04475016).

## 5. Antibody-Drug Conjugates

Cell surface biomarkers can serve as drug targets for monoclonal antibodies or small-molecule inhibitors. Antibody-drug conjugates (ADCs) have the capability to target biomarker-expressing cells irrespective of the biomarker’s functional significance. Examples of successful ADCs include those targeting nectin-4 in urothelial carcinoma and targeting trophoblast cell-surface antigen 2 (TROP-2) in breast cancer and urothelial carcinoma [41,42,43]. Grass et al. conducted an analysis of cell surface protein expression in penile cancer cell lines and patient-derived samples [44]. They analyzed RNA extracted from patient samples (transcriptome) and ribosome-bound RNA from five cell lines (translatome) cross-referenced with a dataset of proteins likely to be localized to the cell membrane based on motif analysis. For patient samples, they also had data on HPV status and survival. Surface expression of putative targets was confirmed with immunohistochemistry. There was robust expression of select targets in the top 85th percentile of expression, and one superfamily of immunoglobulin was prognostic of survival. Receptors accounted for 25% of candidate targets. Transporters accounted for 18%, and showed some differences in expression based on HPV status.

Nectin-4 was not among the top 25% of surface proteins identified by Grass et al.; however, the same group previously characterized nectin-4 specifically. Active transcription of *Nectin-4* was detected in the majority of primary tumors tested, including both high-risk HPV-related and unrelated tumors [45]. Expression was four-fold higher than normal glans tissue in 49% of samples tested. Confirmatory immunohistochemistry demonstrated nectin-4 protein in tumor cells, with 61.4% having moderate or strong staining and 89.4% having >25% of tumor cells positive. 

These results suggest that nectin-4 is a potential target for therapy in patients with metastatic PSCC. A phase II clinical trial of enfortumab vedotin for patients with metastatic or unresectable PSCC is currently in progress (NCT06104618).

Sacituzumab govitecan is another ADC that is approved for treatment of advanced urothelial carcinoma and triple-negative breast cancer [42,43]. It targets cells with expression of the cell surface protein TROP-2. Tekin et al. conducted a study of TROP-2, nectin-4, and PD-L1 expression in PSCC primary tumor samples [14]. Using archived specimens (N = 121) from total or partial penectomies, immunohistochemistry was performed to determine H-score (range, 0 to 300), then expression was classified as low, intermediate, or high for TROP-2 and nectin-4, and positive or negative for PD-L1. An H-score 100–200 for TROP-2 or nectin-4 was considered to be intermediate and >200 was considered to be high; for PD-L1, an H-score >0 was positive. In addition, 37 cases (30.6%) were determined to be high-risk HPV-related. TROP-2 expression was high in 80.7% of cases, which included nearly all (97.3%) of the HPV-related cases and 73.2% of HPV-unrelated cases. Nectin-4 expression was intermediate or high in 79% of cases (Figure 1). PD-L1 expression was detected in 35 of 107 cases tested (32.7%) and was moderately correlated with lower nectin-4 H-scores (median 120 versus 140). Expression of TROP-2 and nectin-4 was predominantly localized to the cell membrane (Figure 2).

These results provide further evidence of robust nectin-4 expression in PSCC, for which therapeutic targeting with enfortumab vedotin is in clinical testing. TROP-2 is also expressed, suggesting a role for ADCs that are directed at TROP-2.

## 6. Combinations with Immunotherapy

As we previously noted, only a small percentage of patients with PSCC are likely to benefit from single-agent therapy with ICIs. A combination of checkpoint inhibitors with other drugs have been successful in multiple cancer types, for example, pembrolizumab plus enfortumab vedotin in advanced urothelial carcinoma [46]. Several phase 1 and phase 2 clinical trials are looking at combinations with checkpoint inhibitors in PSCC (Table 3).

The triplet of ipilimumab, nivolumab, and cabozantinib was tested in phase 1 and phase 2 trials for multiple genitourinary cancer types, including PSCC [47]. In the phase 1 trial, there were nine patients with metastatic PSCC, of which four patients (44.4%) had a partial response. The subsequent phase 2 basket trial includes a cohort of patients with metastatic or unresectable PSCC, with results yet to be reported.

The combination of an HPV vaccine with an ICI has been tested for high-risk HPV-related squamous cell carcinoma from multiple sites. ISA101 is a therapeutic vaccine that targets HPV-specific oncoproteins E6 and E7, inducing expansion of HPV targeted T-cells. In a phase 2 study of ISA101 plus cemiplimab for advanced HPV-16 positive cancer, mostly oropharyngeal, there were responses in 8 of 24 patients (33.3%), and 5 patients had durable response [48]. In another study of ISA101 plus nivolumab in 24 patients with HPV-16-positive cancer, mostly oropharyngeal, there were 8 responses (33.3%), with a median duration of response of 11.2 months and 3 patients without progression at 3 years [49].

An ongoing clinical trial, DUET-4, combines XmAb22841 and pembrolizumab (NCT03849469). XmAb22841 is a bispecific antibody that targets CTLA-4 and LAG-3. DUET-4 is a phase 1 study in advanced solid tumors, including PSCC.

## 7. Conclusions

From laboratory studies, there has emerged a new understanding of PSCC genetic alterations, surfaceome proteins, and the tumor microenvironment. The stage is set to harness these insights for the development of improved therapeutic strategies for patients afflicted with the disease. As we have reviewed here, the most recent experience from clinical trials, unfortunately, reflects the limited efficacy of established chemotherapy options, ICI therapy, or EGFR pathway targeted drugs given individually. The finding of nectin-4 and TROP-2 surface proteins in PSCC offers new targets that must be exploited, with at least one prospective trial of enfortumab vedotin already in progress. Multiple trials of drug combinations with ICI immunotherapy are either recently completed, ongoing, or planned. We conclude that these laboratory and clinical collaborations will ultimately build an improved therapeutic landscape for patients with advanced PSCC. 

## Figures and Tables

**Figure 1 cancers-16-02086-f001:**
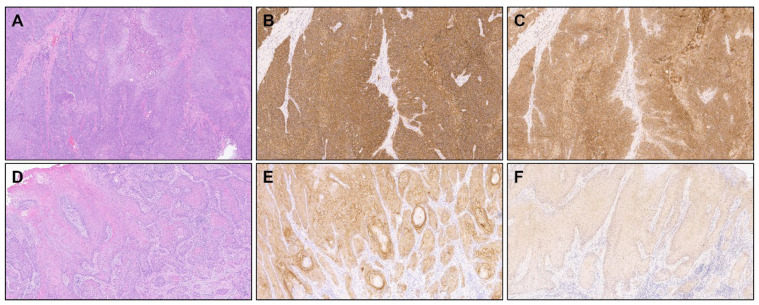
Representative images of penile squamous cell carcinomas with different intensities of TROP-2 and nectin-4 expression using immunohistochemistry. Panels (**A**–**C**) are H&E, TROP-2, and nectin-4 stained images from the same case. This case was assigned TROP-2 and nectin-4 H-scores of 290 and 280, respectively, corresponding to a high expression. Panels (**D**–**F**) are H&E, TROP-2, and nectin-4 stained images from a separate case. This case was assigned TROP-2 and nectin-4 H-scores of 170 and 60, respectively. H-scores are based on the staining intensity multiplied by the percentage of immunoreactive cells and range from 0 to 300. All images were acquired at 100× magnification.

**Figure 2 cancers-16-02086-f002:**
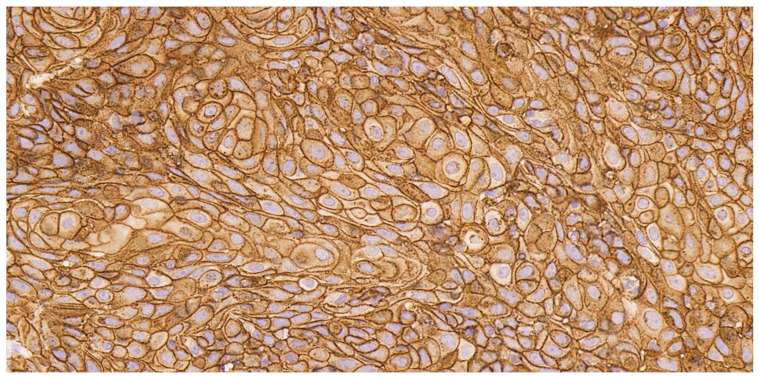
High-power photomicrograph (captured at 400× magnification) demonstrating predominantly membranous TROP-2 staining.

**Table 1 cancers-16-02086-t001:** The most common genomic alterations in PSCC.

	Percent			
	Nazha et al. (2023) [7]	Jacob et al. (2019) [11]	Ali et al. (2016) [12]	McDaniel et al. (2015) [13]
*TP53*	45.5	58	65	45
*CDKN2A*	25.6	47	40	54
*PIK3CA*	24.8		25	20
*TERT*	22.2	45		
*KMT2C*	15.9			
*NOTCH1*	14.1	22	25	9
*KMT2D*	13.6			

**Table 2 cancers-16-02086-t002:** Published studies of EGFR-targeted systemic therapy in PSCC.

Author (Year)	Treatment	N	Comment
Retrospective			
Brown (2014) [35]	Cetuximab	2	1 partial response
	Panitumumab + CTX	1	Partial response
Carthon (2014) [36]	Cetuximab + CTX	16	5 partial responses, 5 stable disease
	Cetuximab	5	1 partial response; 1 stable disease
	Erlotinib	2	Progression as best response
	Gefitinib	1	Progression as best response
Prospective			
Necchi (2016) [37]	Panitumumab (pilot series)	11	ORR 27%
Necchi (2018) [38]	Dacomitinib (phase 2)	28	ORR 32%; PI3K/mTOR pathway gene mutations more common for responders (43% versus 8%)

Abbreviations: CTX, chemotherapy; ORR, overall response rate.

**Table 3 cancers-16-02086-t003:** Clinical trials investigating drug combinations with immune checkpoint inhibitors in PSCC or HPV16+ cancers.

Status	Agent(s)	Tumors	N	ID
Completed	Cisplatin-based chemotherapy + pembrolizumab (HERCULES)	Penile	33	NCT04224740
Recruiting	Cisplatin-based chemotherapy + cemiplimab (EPIC)	Penile	29	ISRCTN95561634
Recruiting *	Cabozantinib + Ipi/Nivo (ICONIC)	Rare GU Tumors	224	NCT03866382
Recruiting	Niraparib + dostarlimab	Penile	25	NCT05526989
Completed	XmAb22841 + pembrolizumab (DUET-4)	Advanced solid tumors	78	NCT03849469
Active, not recruiting	HPV-specific T cells + nivolumab (HESTIA)	HPV-associated cancers	32	NCT02379520
Completed	ISA101b + cemiplimab	Oropharyngeal HPV16+	26	NCT04398524

* Penile cancer cohort has closed. Abbreviations: Ipi/Nivo, ipilimumab, and nivolumab.

## Data Availability

No new data were created or analyzed in this study. Data sharing is not applicable to this article.

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
