# Peer review of "Therapeutic Targets in Advanced Penile Cancer: From Bench to Bedside"

_cancers, 2024, doi:10.3390/cancers16112086_

Round 1
Reviewer 1 Report
Comments and Suggestions for Authors
The article is described in detail and contains a lot of review material, including discussion of significant molecular cascades and pathways that are blocked when using targeted drugs. However, the introduction to the article is rather sparse, and there is not enough information about the main problem. This makes the abstract very general, and the conclusion lacks practical significance .
Comments on the Quality of English Language
The article is described in detail and contains a lot of review material, including discussion of significant molecular cascades and pathways that are blocked when using targeted drugs. However, the introduction to the article is rather sparse, and there is not enough information about the main problem. This makes the abstract very general, and the conclusion lacks practical significance .
Author Response
We thank the reviewer for the comments. In response, the abstract and introduction have been revised to provide more specificity about the purpose and findings of our paper. The conclusion paragraph has been completely rewritten.
Reviewer 2 Report
Comments and Suggestions for Authors
Penile cancer is an uncommon disease. The authors first classify PSCC into two categories, that is, HPV-related and unrelated. Next, in relationship to these two categories, they provide a good summary of current basic research based on genomic profiling. The focus of the manuscript is on a comprehensive update of the clinical research on this rare disease. At this moment, nothing is yet conclusive about how to treat PSCC. However, the review offers a very good snapshot to guide future bench research and clinical trials in this developing medical area. Therefore, it matches the theme of the special issue. The paper is well written and organized. The conclusions are consistent with the presented evidence.
Author Response
We thank the reviewer for the kind remarks.